# Factors Affecting Attitudes towards Older People in Undergraduate Nursing Students

**DOI:** 10.3390/healthcare9091231

**Published:** 2021-09-18

**Authors:** Lourdes López-Hernández, Francisco Miguel Martínez-Arnau, Elena Castellano-Rioja, Marta Botella-Navas, Pilar Pérez-Ros

**Affiliations:** 1Department of Nursing, Universidad Católica de Valencia San Vicente Mártir, 46007 Valencia, Spain; Lourdes.lopez@ucv.es (L.L.-H.); elena.castellano@ucv.es (E.C.-R.); marta.botella@ucv.es (M.B.-N.); 2Department of Physiotherapy, Universitat de València, 46010 Valencia, Spain; 3Frailty and Cognitive Impairment Research Group (FROG), Universitat de València, 46010 Valencia, Spain; maria.p.perez-ros@uv.es; 4Department of Nursing, Faculty of Nursing and Podiatry, Universitat de València, 46010 Valencia, Spain

**Keywords:** ageism, aged, factors, attitude, nursing students

## Abstract

Background: The population of older people is increasing worldwide. The social and healthcare systems need many nurses to care for the elderly. Positive attitudes increase the preference to work with older people and improve the quality of care. This study describes attitudes towards the elderly in a sample of nursing students, and analyzes the potential factors influencing these attitudes. Methods: A cross-sectional study was carried out in nursing students during the academic course 2017–2018. Kogan’s Attitude Toward Old People Scale was used to assess student attitudes towards older people. Results: The study included 377 undergraduate nursing students, of which 75.9% were women. The mean age was 22.23 (5.69) years. Attitude proved positive, with a mean Kogan’s score of 131.04 (12.66). Women had higher scores than men, with a mean difference of 7.76 (95% CI: 4.87–10.66; *p* < 0.001). The male sex, age ≥ 25 years, and previous experience with institutionalized older adults worsened attitudes, while studying the subject of geriatrics, each higher course within the degree, work placements in hospitals and nursing homes, and previous experience with community older adults or with older relatives favored a more positive attitude. Participants with no interest in working with older adults yielded lower scores. Conclusions: Attitudes towards the elderly among nursing students are positive. Women have a more positive attitude. Analyzing the factors that improve attitudes in nursing students is suggested, as it may contribute to improve nursing care.

## 1. Introduction

The proportion of older people has been increasing worldwide for decades [1,2,3]. Europe in particular is facing major demographic changes characterized by a decrease in birth rates and an aging population [4].

Advances in medicine have contributed to a chronic course of different diseases, based on new treatments and managements, with a consequent increase in life expectancy. However, this increase is not directly proportional to an increase in quality of life among older people. Although most older people age satisfactorily, they are more susceptible to declining health and increased dependence [5]. This longer life expectancy has increased the risk of disease, disability, dementia, dysfunction, and dependence [6]. The most widespread stereotypes associate aging with dependency, which can lead to increased negative attitudes towards older people [7].

At the same time, the increase in life expectancy and in the number of older dependent individuals has modified family structures and the provision of care, requiring the help of specialized professionals outside the family environment [8]. The older population is steadily increasing its use of the healthcare system, having become the leading user of social and healthcare resources [9]. Nurses are a key element in the care of older people in all areas. In Spain, 4.1% of the elderly are nursing home residents [8], and in inpatient units, the mean age of patients is 60 years [10]. The Spanish health and social health systems therefore need a large number of nurses working in the care of older people, and they must be prepared to provide quality care. The most suitable environment for acquiring these competences is at university, though they should also be reinforced and updated over the course of professional life [11]. The competencies to be acquired by a future Registered Nurse include knowledge of the aging process and the care of older persons [12], though on the other hand not all nursing degree curricula are the same, and differences are found in Europe and even between universities within the same country. These aspects, in addition to other cultural factors, could interfere in the attitude towards older people, affecting the desire to work in geriatric areas [13,14,15]. A positive or negative attitude towards older people not only conditions the choice of work, but also the quality of the care provided [3,16,17].

A negative attitude towards older people is known as ageism, a term introduced by Butler more than 50 years ago. He considered it as a kind of bigotry similar to racism or sexism, focused in this case on the elderly. “Ageism reflects a deep seated uneasiness on the part of the young and middle-aged—a personal revulsion to and distaste for growing old, disease, disability; and fear of powerlessness, uselessness, and death”.

Age discrimination leads younger people to distance themselves from those who are older to the point where they even cease to identify older individuals as human beings [18].

Despite geronto-geriatric instruction in university education, ageism is also present among nursing professionals. There are two theories that can be used to explain ageism in nursing: social identity theory and terror management theory. The social identity theory refers to the fact that young people, in order to maintain and/or increase their self-esteem, identify themselves better with people of their own age, distancing themselves from the rest, and thus valuing being a member of the preferred social group. In this way, the younger group will be valued positively, and the older group will be valued negatively. The terror management theory in turn refers to young people avoiding contact with things that frighten them, death being one of them. Older people would remind them of the end of life and therefore of the closeness of death—these aspects being viewed as negative. Experiences with older people provoke feelings of vulnerability and offer disconcerting reminders of the fragility of human life. Other authors have also identified that changes in physical appearance may encourage ageist attitudes in younger people, as older persons no longer present attractive features [19]

Several authors have researched this issue in nursing students; Kydd et al. [15] observed students in Scotland had a better attitude towards the elderly than Swedish or American students. They highlighted working conditions as the main reason for refusing to work with the elderly. Lee [20] reported differences in attitude between different ethnic groups, with a lesser negative attitude towards the elderly being seen in students of Asian origin than in Caucasian participants. Other studies have observed a relationship between the more advanced courses of the degree and more positive attitudes towards the elderly in nursing students [21,22]. Thus, the curriculum of the degree could influence student attitudes [21]. An increase in knowledge about the aging process can contribute to positive attitudes towards older people [23,24], while a lower level of knowledge might result in a more negative attitude [25].

Not only is knowledge about the aging process and care related to attitudes, but there are certain other non-modifiable factors, such as age or sex, as well as modifiable factors, such as work placements or previous experiences with older people, that could modulate attitudes towards such individuals [26,27].

The present study explores the attitudes towards older people among nursing students, seeking to identify the factors that can positively or negatively influence such attitudes.

## 2. Materials and Methods

### 2.1. Design and Sample

A cross-sectional study was carried out, involving nursing degree students (Universidad Católica de Valencia San Vicente Mártir, Valencia, Spain), during the academic course 2017–2018. The inclusion criteria were enrollment in any course of the Nursing School, age 18 ≥ years, and agreement to participate in the study. Students failing to sign the informed consent form were excluded. Nursing studies at the Universidad Católica de Valencia San Vicente Mártir cover four academic years [28] (Figure 1). The syllabus distribution is based on the recommendations of the Spanish Department of Education and the European Higher Education Area (EHEA). The first year comprises 90% of basic modules in Health Sciences such as anatomy, physiology, etc., in line with other degrees in Health Sciences. The second year comprises 50% of subjects in basic Health Sciences and the rest of subjects specific to the nursing discipline, such as Care of Adults and Care of Children. In addition, there are two work placement rotations where students complete a 200-h work placement in a hospital and a 120-h work placement in a nursing home. In the third year, the students complete nursing care subjects, and specifically Care of the Elderly. In addition, they complete three work placement rotations: two of them in hospitals (one covering 120 h and the second 200 h) and one (200 h) in nursing homes. In the fourth year, the students receive nursing care subjects (emergency and mental health care), complete an undergraduate thesis, and undergo the last work placement rotations (each with a duration of 400 h): one of them in a hospital (Intensive Care Unit or emergency service) and the other in the primary care setting.

### 2.2. Sample Size

The sample size was calculated using XLstats^®^ v. 23 (Addinsoft, Paris, France), and based on a previous study, where the KAOP scores in nursing students were 136.26 (standard deviation [SD]: 17.47) [29]. Adopting a sampling error of ±3% and α = 5, and with an expected loss rate of 5%, the minimum sample required for the study was found to be 310 nursing students.

A total of 377 nursing students were recruited; no one refused to participate, and no surveys had to be withdrawn due to errors in completion.

### 2.3. Instruments

The instrument selected for data collection was Kogan’s Attitude Toward Old People Scale, which was used to measure attitudes in the selected sample. This instrument consists of 34 items, 17 of which are positive and 17 negative in terms of attitudes towards the elderly, scored according to a Likert scale, with 1 being the lowest agreement and 6 the highest agreement. The negative items were inverted for final calculation of the values obtained on the scale, so that the final score of the instrument indicated that the higher the value, the lower the negative attitude and vice versa. Cronbach’s alpha was found to be 0.73 and 0.83 for positive and negative scales, respectively [30]; Hilt evaluated the scale as a whole and found its Cronbach’s alpha to be 0.81 [31]. In addition, the participants were asked about their age and sex, whether they had studied nursing care with the older patient subject in mind, whether they had completed a practical period in nursing homes or hospitals, what kind of contact they had with older people in their lives, and whether they would like to work with older people in the future.

### 2.4. Procedure

Data collection took place in the first week of February 2018. The distribution of students was a maximum of 60 in each group, with 12 groups in total—four for each academic year. Two people from the research team contacted the lecturers of each group beforehand so that they would allow them to enter the classroom and give them a maximum of 30 min. A booklet containing the information sheet, the informed consent form and the variables collection sheet, together with Kogan’s scale, was handed out in paper format. Approximately 10 min were spent explaining the reason for the study and signing the informed consent form, followed by a maximum of 20 min for the completion of Kogan’s scale and the rest of the variables.

### 2.5. Statistical Analysis

Variables were reported as proportions and/or the mean and standard deviation (SD). Parametric tests (Student *t*-test) were used to compare continuous variables, while nonparametric tests (chi-squared test and linear trend test) were used to compare categorical variables. In addition, the validity of the instrument was analyzed using Cronbach’s alpha.

The Pearson correlation coefficient was used to correlate Kogan’s scale and quantitative variables, and a linear regression model was constructed to assess the importance of the different factors related to Kogan’s scores. We considered the complete model with all the variables, as the authors wanted to know whether it exerts a positive or negative influence upon Kogan’s total score.

The study data were entered in MS Excel spreadsheets, and the statistical analysis was performed using the SPSS statistical package (IBM Corp. Released 2010. IBM SPSS Statistics for Windows, Version 23.0. IBM Corp, Armonk, NY, USA).

## 3. Results

A total of 377 participants were included in the study. Most belonged to the first year (51.2%, *n* = 193). The mean age was 22.23 (5.69) years, with a female predominance. A total of 20.2% of the students (*n* = 76) were over 25 years of age. Most of the students had not completed the practicum rotations and had not yet taken the course Care of the Elderly. Contact with older people was mainly limited to relatives, and only 30.2% of the students considered working with older people as a valid option. The mean Kogan’s score was 131.04 (12.66) out of 204, indicating a positive attitude (defined as > 102 points), but far from the highest possible score (Table 1). Cronbach’s alpha was 0.71.

The scores of Kogan’s scale were analyzed in relation to the other variables. No correlation was found between Kogan’s score and age (r = −0.05; *p* = 0.29) or with the positive and negative items (r = −0.03; *p* = 0.62 and r = −0.06; *p* = 0.25, respectively).

Females scored higher than males (mean diff. = 7.76; 95%CI: 4.87–10.66; *p* < 0.001). No differences were found between courses, nor after work placement, nor after studying Care of the Elderly. Students who had been in contact with institutionalized older adults or who had no previous experience with older people had lower and therefore more negative scores than the rest, though without statistically significant differences. (Table 2).

Participants who did not want to work with older adults or did not consider this possibility unless they found it difficult to find other work yielded lower scores than those who did consider working with older adults at the end of their studies (Table 2).

The variables were included in a linear regression model, yielding a statistically significant model (F = 3.78, *p* < 0.001; R = 0.275, R2 = 0.076). The male sex, age ≥ 25 years, and previous experience caring for the elderly decreased positive attitudes towards older people, whereas studying Care of the Elderly and enrollment in more advanced nursing courses increased positive attitudes. Contact with the elderly both in the OCA setting and with relatives improved positive attitudes the most (Table 3).

## 4. Discussion

Ageism is present in our society, and in 2018 the World Health Organization (WHO) even launched a proposal to work against ageism on a global level [32]. Negative attitudes towards older people in healthcare settings can also impair the quality of care [33]. It is necessary to know the attitude of nursing students who are going to take care of this population group in the near future [34,35,36]. The aim of the present study was to evaluate the attitude and the factors that can influence nursing students both positively and negatively. The scores obtained in the assessment of attitude based on Kogan’s scale indicated a positive attitude towards older people [30], with the female sex, better knowledge and previous experiences being identified as the factors that positively influence more positive attitudes.

The few studies that have been published on nursing students in Spanish universities provide data indicating slightly positive scores [37,38]. The scores obtained were similar to those obtained in other studies on nursing students in Europe and the United States [39,40,41], and slightly higher than in Asian students [42]. The cultural background seems to be an important factor, as other studies have also reported more negative scores in Asian countries compared to western countries [43].

In order to propose interventions to improve attitudes towards older people, it is necessary to know which factors—apart from cultural aspects—are involved in generating positive or negative attitudes among students. Gender is the main factor, and in our study there was a more positive attitude among women. There is evidence of a more ageist attitude in men than in women [44,45], though it is also true that this trend is less pronounced in younger populations [46]. However, the literature is not consistent in this regard, yielding inconclusive results [47,48] or a more positive attitude in males [49,50]. These results should be analyzed in relation to different cultural aspects.

Our results indicate more negative attitudes in people over 25 years of age, though without reaching statistical significance. More in-depth studies should be carried out, as these results could be attributable to generational cultural aspects [51]. In addition, nursing students are characterized by the presence of a large group of nursing assistants with a higher mean age than those coming from secondary education, and they may have previous experience of working with older people. This could bias the results and represents an aspect to be analyzed in future studies.

Enrollment in the more advanced courses of the degree had a positive impact upon the improvement of attitudes in our sample. There is disagreement in the literature in this respect, with some studies indicating that attitudes improve with knowledge and contact with older people throughout the university career, while other studies indicate that the opposite may be true after negative experiences during work placements, which in turn influence future attitudes in undergraduate studies [52,53]. This same discrepancy has been found in studies of medical students, subsequently affecting the choice of medical specialization in geriatrics [54,55].

Studying a subject on “Caring for the Elderly” should improve attitudes. Obviously, knowledge helps to understand the aging process and therefore to understand certain disease conditions, which are more prevalent in older people. It would also help to distinguish between what is related to the physiological ageing process itself and what is not, in order to avoid negative stereotypes [34], though knowledge does not always change attitudes in nursing students [56].

What does seem to exert a major influence is the previous experience which students have with older people. Previous experiences in working with older institutionalized adults tend to decrease positive attitudes, while experiences with older community adults and older relatives tend to improve attitudes [57,58,59]. Similar results were obtained in the study by Dahlke, indicating that attitudes are influenced by the type of previous experience the students had, and that cultural aspects of caring for older relatives should also be taken into account [58,59]. In addition to previous experiences in work placements, the perception of aging in the work placement setting is important, as well as the knowledge and attitude of the work practice supervisor. The skills and knowledge of lecturers in the classroom and at the work placement centers are also important for the perception of aging in the placement settings [60,61].

Students with poorer attitudes also did not consider working with older people or did not have such work as their first choice. Attitudes are directly related to work preference [62]. Most of the people cared for in the health and social care systems are older people (community, hospital, and socio-health care); therefore, unless work focuses exclusively on pediatric or young patients, nurses will inevitably work with older people. Nursing professionals are indispensable for the care of the elderly in all areas and, as mentioned, a positive attitude is directly related to quality of care. Universities should consider the analysis of attitudes and review their syllabus, placement settings, and practice supervisors and the rest of the health professionals involved and, if needed, implement programs to improve attitudes towards older people among the healthcare professionals [63,64].

The limitations of the present study include its cross-sectional nature. Analysis according to the type of work placement involved has not been possible, nor has it been possible to collect data from fourth year students, as they do not attend Nursing School at this time, due to their work placements. The analysis of the data was carried out prior to the COVID-19 pandemic and has highlighted the large number of older people with a greater susceptibility to vulnerability. It would be useful to conduct further studies to determine whether there have been attitudinal changes after the pandemic, or whether these factors have been modified.

## 5. Conclusions

Attitudes towards older people among nursing students are positive. Women have a more positive attitude, and there are factors that could contribute to increase positive attitudes, such as studying a subject related to the care older people, or having previous personal experience with older relatives. Longitudinal studies are needed to determine which factors directly influence attitudes throughout the university career.

## Figures and Tables

**Figure 1 healthcare-09-01231-f001:**
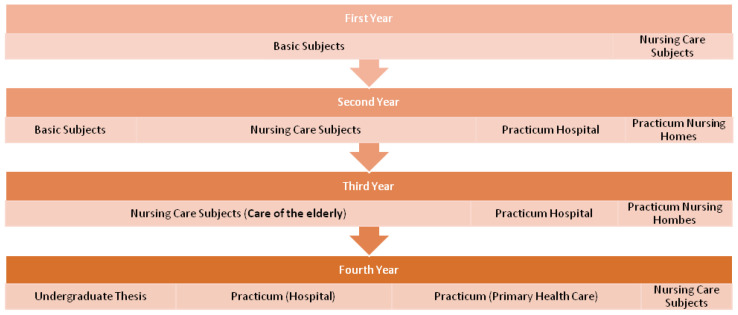
Syllabus of the Degree in Nursing 2010.

**Table 1 healthcare-09-01231-t001:** Characteristics of the study sample.

	Mean (SD)/*n* (%)
Age	22.23 (5.69)
Sex	
Male	91 (24.1)
Female	286 (75.9)
Academic course	
First	193 (51.2)
Second	95 (25.2)
Third	89 (23.6)
Have you done work placements in hospitals or nursing homes? (yes)	107 (28.4)
Have you studied Care of the Elderly? (yes)	91 (24.1)
Do you have previous experience caring for relatives or living with older people?	
None	51 (13.5)
Previous experience with OIA	60 (15.9)
Previous experience with OCA	84 (22.3)
Previous experience with older relatives at home	182 (48.3)
Would you like to work with older people after your studies?	
Yes	114 (30.2)
No	52 (13.8)
Only if I do not find work in another area	211 (56.0)
Kogan’s total	131.04 (12.66)
Kogan’s negative items	69.43 (9.05)
Kogan’s positive items	61.6 (7.84)
Kogan’s ranges	
More negatives	0 (0)
Negatives	3 (0.8)
Neutrals	265 (70.3)
Positives	108 (28.6)
More positives	1 (0.3)

OCA: older community adults; OIA: older institutionalized adults.

**Table 2 healthcare-09-01231-t002:** Mean and SD of Kogan’s scores according to different variables.

	Mean Total Scores (SD)/ Significant at		Negative Items Scores (SD)/ Significant at		Positive Items Scores (SD)/ Significant at	
Sex						
Female	132.91 (12.44)	*p* < 0.001	70.55 (8.73)	*p* < 0.001	62.36 (7.92)	*p* = 0.001
Male	125.14 (11.53)		65.91(9.19)		59.23(7.15)	
Academic course						
First	130.76 (13.29)	*p* = 0.769	68.70 (9.26)	*p* = 0.205	62.05 (7.73)	*p* = 0.527
Second	131.86 (12.96)		70.70(9.08)		61.15 (9.06)	
Third	130.77 (10.92)		69.65 (8.51)		61.12 (6.64)	
Have you done work placements in hospitals or nursing homes?						
Yes	130.40 (11.65)	*p* = 0.475	69.31 (8.61)	*p* = 0.829	61.08 (6.96)	*p* = 0.392
No	131.38 (12.97)		69.53(9.21)		61.85(8.16)	
Have you studied Care of the Elderly?						
Yes	130.50 (11.17)	*p* = 0.601	69.38 (8.85)	*p* = 0.908	61.12 (6.60)	*p* = 0.429
No	131.30 (13.04)		69.50 (9.10)		61.79 (8.19)	
Do you have previous experience caring for relatives or living with older people?						
None	129.20 (12.54)	*p* = 0.304	68.63 (9.97)	*p* = 0.816	60.57 (6.85)	*p* = 0.259
Previous experience with OIA	129.62 (12.38)		69.15 (8.98)		60.47 (7.63)	
Previous experience with OCA	132.88 (12.50)		70.12 (8.64)		62.76 (8.46)	
Previous experience with older relatives at home	131.17 (12.83)		69.43 (9.05)		61.73 (7.86)	
Would you like to work with older people after your studies?						
Yes	133.31 (12.91)	*p* = 0.064	70.50 (8.83)	*p* = 0.137	62.82 (8.26)	*p* = 0.103
No	129.36 (12.28)		67.5 (9.44)		61.87 (7.96)	
Only if I do not find work in another area	130.22 (12.26)		69.33 (9.04)		60.89 (7.53)	

OCA: Older Community adults; OIA: Older institutionalized adults; SD: standard deviation.

**Table 3 healthcare-09-01231-t003:** Linear regression model of the total Kogan’s scores.

			95% CI for B		
	B	Std. Error	Lower Limit	Upper Limit	Sig.
Kogan’s scores	130.71	3.64	123.55	137.86	<0.001
Male	−7.56	1.52	−10.55	−4.56	<0.001
Age ≥ 25 years	−0.49	1.65	−3.74	2.76	0.767
Previous experience with OIA	−0.74	2.52	−5.69	4.22	0.770
Academic course	0.85	1.52	−2.14	3.84	0.575
Geriatrics subject	0.49	4.31	−7.99	8.97	0.909
Practicum in hospitals and nursing homes	1.22	3.22	−5.10	7.54	0.705
Previous experience with relatives	1.86	2.23	−2.53	6.24	0.406
Previous experience with OCA	0.19	1.99	−3.72	4.09	0.925

OCA: older community adults; OIA: older institutionalized adults.

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
