# Peer review of "Factors Affecting Attitudes towards Older People in Undergraduate Nursing Students"

_healthcare, 2021, doi:10.3390/healthcare9091231_

Round 1

Reviewer 1 Report

The phrase beginning at l. 176 (Likewise...) could be clearer. Please elaborate your point.

Idem at l. 241: 'impair attitudes' In which specific ways attitudes are impaired?

This is by all means an impeccable piece of work. The English, presentation, argumentation and conclusions are reproachless. Nevertheless, it provides little advances in scholarship and the results are what could have been expected.

Reviewer 2 Report

Thank you for reviewing this paper. I have just two points of comments.

First of all, the title should be changed to "Factors influencing (or affecting) attitudes towards older people in undergraduate nursing students." You didn't verify positive factors, but just conducted to explore factors influencing attitudes. 

In Abstract, the conclusion is better to show the suggestion like conclusion part of this manuscript. For example, the expression of '~ is mandatory ~' is not good. I think '~ is suggested~' is better.

Good Luck.  

Reviewer 3 Report

The authors present us with an interesting study in the field of geriatrics and future professionals, however, I have some doubts and a number of questions are advised to improve the manuscript.

The main problem I see is that the authors refer to outdated and old data, why publish this data now? The study is of the perception of students in the 2017-18 academic year, that perception is not current, it may be that this perception has changed, if the contents of the subjects, teachers who teach these subjects, the situation of covid19... from 2017 to 2021 have been modified.

This issue should be addressed in depth in the discussion.

Introduction,

The subject of the study should be further contextualised.

Methodology

The data collection procedure is not clear. How did the students fill in the surveys? Online? Did they meet and fill them in on site? Depending on how these data were collected, there may be a number of limitations or biases that should be discussed in the discussion section.

Discussion

Have the authors reviewed the curricula of the same subjects with other Spanish universities? Are there differences with respect to their curricula to see if the perception or results can be influenced?

Round 2

Reviewer 3 Report

The authors have answered and completed all the suggested questions.